# Structural foundation for the role of enterococcal PrgB in conjugation, biofilm formation, and virulence

**Wei-Sheng Sun[1,2], Lena Lassinantti[1], Michael Järvå[1], Andreas Schmitt[1], Josy ter Beek[1,2], Ronnie P-A Berntsson[1,2]\***

[1]Department of Medical Biochemistry and Biophysics, Umeå University, Umeå, Sweden; [2]Wallenberg Centre for Molecular Medicine, Umeå University, Umeå, Sweden

**\*For correspondence:**
ronnie.berntsson@umu.se

**Competing interest:** The authors declare that no competing interests exist.

**Abstract** Type 4 Secretion Systems are a main driver for the spread of antibiotic resistance genes and virulence factors in bacteria. In Gram-positives, these secretion systems often rely on surface adhesins to enhance cellular aggregation and mating-pair formation. One of the best studied adhesins is PrgB from the conjugative plasmid pCF10 of *Enterococcus faecalis*, which has been shown to play major roles in conjugation, biofilm formation, and importantly also in bacterial virulence. Since *prgB* orthologs exist on a large number of conjugative plasmids in various different species, this makes PrgB a model protein for this widespread virulence factor. After characterizing the polymer adhesin domain of PrgB previously, we here report the structure for almost the entire remainder of PrgB, which reveals that PrgB contains four immunoglobulin (Ig)-like domains. Based on this new insight, we re-evaluate previously studied variants and present new in vivo data where specific domains or conserved residues have been removed. For the first time, we can show a decoupling of cellular aggregation from biofilm formation and conjugation in *prgB* mutant phenotypes. Based on the presented data, we propose a new functional model to explain how PrgB mediates its different functions. We hypothesize that the Ig-like domains act as a rigid stalk that presents the polymer adhesin domain at the right distance from the cell wall.

## eLife assessment

This study presents **valuable** structural data for the bacterial adhesin PrgB, an atypical microbial cell surface-anchored polypeptide that binds DNA. There is **convincing** support for the claims regarding the overall function and importance of individual domains, which integrate a wide range of new and previously published experimental data. The structure-based model of PrgB molecular activity will be impactful in the field of bacterial adhesins, conjugation, and biofilm formation, especially because it focuses on a clinically-important Gram-positive pathogen, whereas most work in the field has been focused on Gram-negative model systems.

## Introduction

*Enterococcus faecalis* is one of the leading causes of hospital acquired infections, such as urinary tract infections and endocarditis (*Gilmore et al., 2013*; *Hidron et al., 2008*). These infections are difficult to treat as *E. faecalis* has the tendency to form biofilms and is often resistant to various antibiotics. They are also notorious for spreading antibiotic resistance and other fitness advantages by transfer of mobile genetic elements, which can be located on conjugative plasmids or in the chromosome (*Palmer et al., 2010*; *Palmer and Gilmore, 2010*). Conjugative plasmids usually also encode a Type

4 Secretion System (T4SS) that mediates its transfer, via conjugation, from a donor cell to a recipient cell (*Trokter and Waksman, 2018*; *Cabezón et al., 2015*; *Koraimann, 2018*; *Grohmann et al., 2018*). However, conjugative plasmids and their T4SS have almost exclusively been studied in Gram-negative model systems (*Grohmann et al., 2018*).

One of the few well-characterized Gram-positive conjugative plasmids is pCF10 from *E. faecalis* (*Cabezón et al., 2015*; *Hirt et al., 2005*; *Dunny and Berntsson, 2016*). This conjugative plasmid contains a ~27-kbp operon that is tightly regulated by the $P_Q$ promoter (*Chandler et al., 2005*; *Dunny, 2013*; *Lassinantti et al., 2021*) and that encodes all proteins needed for conjugation. This operon also encodes three cell wall anchored proteins: PrgA, PrgB, and PrgC. PrgA is a conjugation regulator that provides surface exclusion to prevent unwanted conjugation. We have previously shown that PrgA consists of a protease domain that is presented far away from the cell wall via a long stalk and that it is likely mediating the proteolytic cleavage of PrgB (*Schmitt et al., 2020*; *Järvå et al., 2020*). PrgC is a virulence factor, but its function and structure remain unknown (*Bhatty et al., 2015*). PrgB is the main adhesin produced by pCF10 and has been studied for well over three decades. This protein is around 140 kDa in size and possesses an N-terminal signal sequence and a C-terminal LPXTG cell wall anchor motif (*Chuang et al., 2009*). PrgB, which is indicated to function as a dimer in vivo (*Schmitt et al., 2018*), distributes over the entire surface of the cell wall and increases cellular aggregation, biofilm formation, and the efficiency of plasmid transfer (*Olmsted et al., 1991*; *Olmsted et al., 1993*). Several mammalian infection model systems have shown that PrgB is a strong virulence factor (*Chuang et al., 2009*; *Süssmuth et al., 2000*; *Schlievert et al., 2010*; *Chuang-Smith et al., 2010*; *Rakita et al., 1999*; *Vanek et al., 1999*; *Hirt et al., 2022*). One reason for this virulence is that PrgB mediates biofilm formation in an extracellular DNA (eDNA)-dependent manner (*Bhatty et al., 2015*). Homologs of

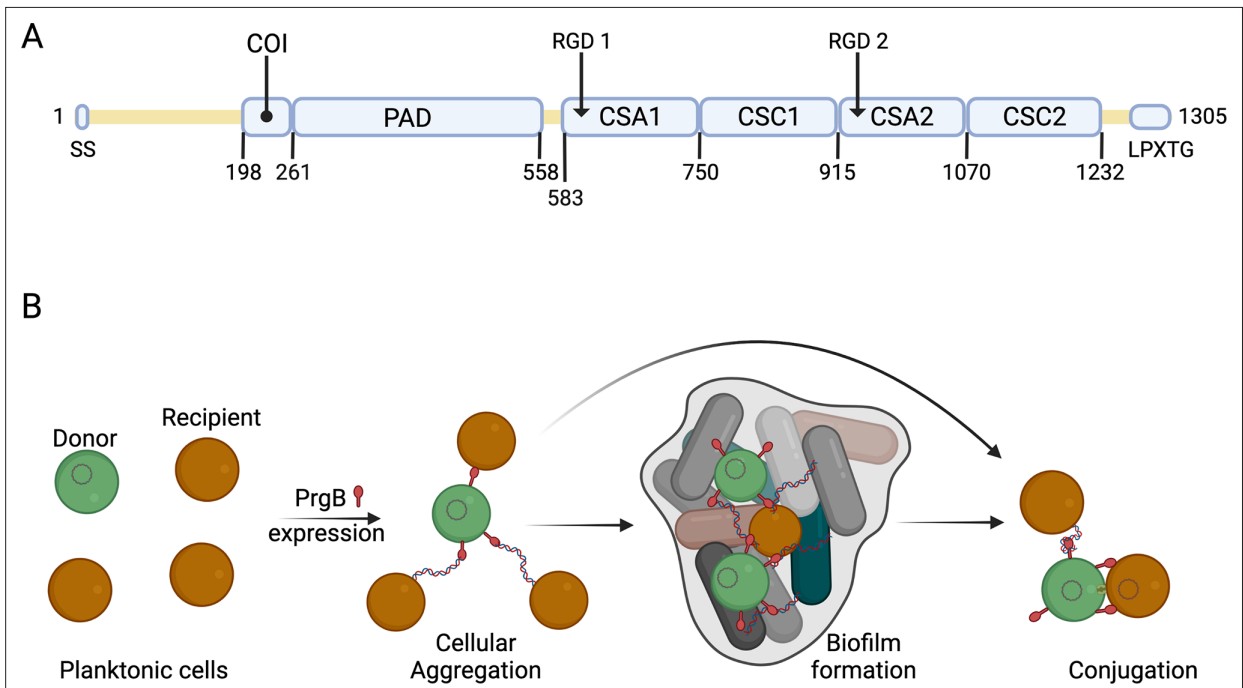

**Figure 1.** Schematic overview of PrgB domain organization and function. (**A**) Updated schematic overview of the domain organization of PrgB. SS: signal sequence, COI: coiled-coil, PAD: polymer adhesin domain, CSA: adhesin isopeptide-forming adherence domain, CSC: cell-surface antigen C-terminal domain, LPXTG: cell wall anchor sequence. The PrgA cleavage site is located between the polymer adhesin domain and the first immunoglobulin (Ig)-like domain, and has the sequence IFNYGNPKEP. (**B**) In a setting with a donor cell (green) and multiple recipient cells (brown), PrgB is produced and sits on the cell wall. There it enhances cellular aggregation and/or biofilm formation, either by directly binding lipoteichoic acid (LTA) from the cell wall of a recipient or by binding first to extracellular DNA (eDNA). PrgB compacts the eDNA, and thereby likely pulls the recipient cells closer. Once close enough, PrgB binds to the LTA of the recipient bacteria and facilitates mating-pair formation and conjugation.

The online version of this article includes the following source data and figure supplement(s) for figure 1:

**Figure supplement 1.** Purification of PrgB.

**Figure supplement 1—source data 1.** Raw image of the SDS-PAGE in *Figure 1—figure supplement 1B*.

PrgB have been identified in many other conjugative plasmids (*Järvå et al., 2020*), suggesting that PrgB-like proteins confer important roles in a large number of bacterial species (*Muscholl et al., 1993*; *Galli et al., 1992*; *Waters and Dunny, 2001*).

PrgB was initially identified as one of the driving forces in cellular aggregation (*Olmsted et al., 1991*). To understand how it mediated this process, previous research has tried to identify the various protein domains that are present in PrgB and evaluate their function(s). Two RGD (Arg–Gly–Asp) motifs were identified (see *Figure 1A*) and found to be important for vegetation and biofilm formation in the host tissue environment (*Chuang et al., 2009*; *Chuang-Smith et al., 2010*). The N-terminal half of PrgB was found to be required for aggregation and to bind lipoteichoic acid (LTA) (*Waters and Dunny, 2001*; *Waters et al., 2003*; *Waters et al., 2004*), which is a major constituent of the cell wall in Gram-positive bacteria. In 2018, we solved the structure of PrgB$_{246-558}$ and showed that it has a lectin-like fold that was most similar to adhesins from various oral Streptococci. As these adhesins are known to bind various polymers, we subsequently referred to PrgB$_{246-558}$ as the polymer adhesin domain (*Järvå et al., 2020*). We have shown that this domain can bind both LTA and eDNA in a competitive manner. Bound eDNA is thereby strongly compacted as it is wrapped around the domain's positively charged surface (*Schmitt et al., 2018*). We therefore proposed that PrgB could use eDNA to promote cell-to-cell contacts, as an alternative to direct binding to the LTA from a recipient cell (*Figure 1B*). As all described polymer adhesin domains, PrgB has a central ridge with a conserved cation-binding site (*Järvå et al., 2020*; *Kohler et al., 2018*). In the homologous GbpC, from *Streptococcus mutans*, this site has been suggested to bind glucans (*Mieher et al., 2018*). However, no interaction with glucans has been observed for PrgB or any other homologs (*Forsgren et al., 2009*). Thus, the importance of this conserved motif remains an open question.

The polymer adhesin domain plays an important role in the function of PrgB, but it only accounts for around a quarter of the entire protein. Here, we present the structure of almost the entire remainder of PrgB. This allows us to put the large amount of available phenotypic data into a structural context and explain a lot of previous observations. We also constructed several new mutants of *prgB* that better fitted the found domain organization and analyzed their in vivo effects on cellular aggregation, biofilm formation, and conjugation efficiency. Based on our findings, we conclude with an updated model of how PrgB mediates its different functions.

## Results

### PrgB$_{584-1233}$ contains four immunoglobulin-like domains

Previous bioinformatics and structural analysis of PrgB proposed that PrgB consisted of three domains; the previously crystallized polymer adhesin domain responsible for eDNA and LTA binding, and two domains with RGD (Arg–Gly–Asp) motifs (*Schmitt et al., 2018*). However, when we reanalyzed the PrgB sequence with the new structure-prediction tools that are available, such as AlphaFold (*Jumper et al., 2021*), it became clear that this model was partially incorrect. PrgB seems to consist of an N-terminal disordered region (residues 35–197), followed by a newly identified coiled-coil (COI) domain (residues 198–257), the previously crystallized polymer adhesin domain (residues 261–558), a linker region (residues 559–582), four immunoglobulin (Ig)-like domains (residues 583–1232), and finally the C-terminal disordered region containing the LPXTG motif (residues 1263–1305) that gets anchored to the cell wall (*Figure 1A*). The Ig-like domains seem to come in pairs of two slightly different structures, denoted as CSA1–CSC1 (first pair) and CSA2–CSC2 (second pair), as named in InterPro (CSA from IPR026345; adhesin isopeptide-forming adherence domain, and CSC from IPR032300; cell-surface antigen C-terminal). To verify this updated domain organization of PrgB, we set out to experimentally determine its structure using a combination of X-ray crystallography and cryo-EM methods.

As described in *Schmitt et al., 2018*, we were not able to produce full-length PrgB in *E. coli*, but instead produced and purified PrgB$_{188-1233}$. This version of the protein only lacks the disordered N-terminal region and the LPXTG anchor and elutes from size exclusion chromatography in two peaks corresponding to dimeric and a monomeric PrgB. PrgB is in a monomer–dimer equilibrium and the dimer has been described as the main biologically functional unit in vivo (*Koraimann, 2018*). The monomeric fraction (*Figure 1—figure supplement 1*) was successfully used for crystallization trials. Crystals belonging to space group P2$_1$2$_1$2$_1$ appeared after 8–12 weeks, diffracted to 1.85 Å and contained two molecules in the asymmetric unit. The crystallographic phase problem was solved using

molecular replacement with SspB (PDB: 2WOY) as a search model. Surprisingly, the resulting electron density lacked the previously crystallized polymer adhesin domain of PrgB, and instead only contained residues 584–1233. There is also no space in the crystal packing to allow for a flexible polymer adhesin domain, so it was likely cleaved off in the crystallization drop before the crystals were formed. The modeled protein indeed consists of four Ig-like domains: two CSA and two CSC domains (*Figure 2A*). Previous bioinformatics analysis showed that various homologous adhesin proteins contain different numbers of Ig-like domains (*Järvå et al., 2020*), but a DALI (*Holm, 2020*) analysis of PrgB$_{584–1233}$ showed that there is no previously solved structure in the PDB that contains four of these Ig-domains coupled together. There are, however, homologous structures available with either two or three Ig-domains. The closest structural homologs are the C-terminal parts of Antigen I/II proteins from oral Streptococci, for example the surface protein AspA from *Streptococcus pyogenes* (*Hall et al., 2014*) (PDB code: 4OFQ), which has three Ig-domains and a root mean square deviation(r.m.s.d.) to the Ig-like domains from PrgB of 3.2 Å over 337 residues, or the BspA protein from *Streptococcus agalactiae* (*Rego et al., 2016*) (PDB code: 4ZLP) which has two Ig-domains and an r.m.s.d. of 3.3 Å over 334 residues (see *Supplementary file 3* for an overview of the highest ranked DALI hits). The r.m.s.d. decreases to ca. 1 Å if the individual Ig-like domains are superimposed upon each other.

In each of the Ig-like domains of PrgB, isopeptide bonds are formed between a lysine and an asparagine, a bond which is further stabilized by an aspartic acid (*Figure 2B*). This feature is also present in various homologous Ig-like domains from Antigen I/II proteins (*Hall et al., 2014*; *Larson et al., 2011*; *Forsgren et al., 2010*). There is density in the conserved metal-binding site of the CSA2 domain that has been suggested to bind Ca$^{2+}$ in the Antigen I/II proteins. Refinement of our structure indicated that Mg$^{2+}$ was the best fit to the density (*Figure 2C*). The homologous C2 domains from AspA (*Hall et al., 2014*), Pas (*Mieher et al., 2021*), SpaP (*Larson et al., 2011*), and SspB (*Forsgren et al., 2010*), each have an extra structural feature termed the BAR (SspB adherence region) domain, which mediates adherence in these proteins. This BAR domain is absent in PrgB (*Figure 2—figure supplement 1*).

As we did not manage to crystallize PrgB$_{188–1233}$ without the loss of the polymer adhesin domain, we tried to determine its structure via cryo-EM and single particle analysis. Two datasets of PrgB, one with and one without ssDNA (120 bases), were collected. This yielded relatively low-resolution volumes (8 and 11 Å, respectively). See *Figure 2—figure supplement 2* for an overview of the processing. Docking in the X-ray structures of the stalk domain (PrgB$_{584–1233}$) and the previously solved polymer adhesin domain (PDB code: 6EVU) (*Schmitt et al., 2018*) into the volumes weakly indicated that the polymer adhesin domain could be interacting with the stalk domain in the absence of DNA (*Figure 2—figure supplement 3*). Therefore, we set out to test whether the polymer adhesin domain binds to the Ig-like domains from the stalk domain (PrgB$_{584–1233}$) in vitro. However, neither size-exclusion chromatography nor native polyacrylamide gel electrophoresis (PAGE) indicated any binding of the polymer adhesin domain to the stalk domain in vitro (*Figure 2—figure supplement 4*).

## In vivo assays

Based on the new structural insights for PrgB, we decided to study the importance of the newly defined COI domain and the Ig-like domains. This was done by complementing *E. faecalis* OG1RF pCF10Δ*prgB* with different *prgB* mutants (from a nisin-inducible plasmid) and characterizing their phenotypes in cellular aggregation, biofilm formation, and plasmid transfer efficiency. In line with previous experiments (*Bhatty et al., 2015*), complementing pCF10Δ*prgB* with exogenous PrgB from the pMSP3545S vector rescues all phenotypes. Aggregation and biofilm formation, both measured after overnight incubation, are even slightly increased as compared to wild-type pCF10 (*Figure 3A, B*, columns 1 and 3), possibly due to a slightly increased production of PrgB (*Figure 3—figure supplement 1*, lanes 1 and 3).

We found that expression of PrgB without the newly identified COI domain could not rescue the aggregation phenotype of the deletion strain (*Figure 3A*, column 4). Deletion of either the CSA1 or the CSC2 domain did not affect PrgB-mediated aggregation (*Figure 3A*, columns 8 and 9). However, complementation with PrgB$_{ΔCSA2–CSC2}$ could only partially rescue the aggregation phenotype of the *E. faecalis* OG1RF pCF10Δ*prgB* strain (*Figure 3A*, column 7) and no rescue at all was seen in the PrgB variants with both the CSA1 and CSC1 domain deleted or without any Ig-like domains (CSA1–CSC1–CSA2–CSC2) (*Figure 3A*, columns 5 and 6).

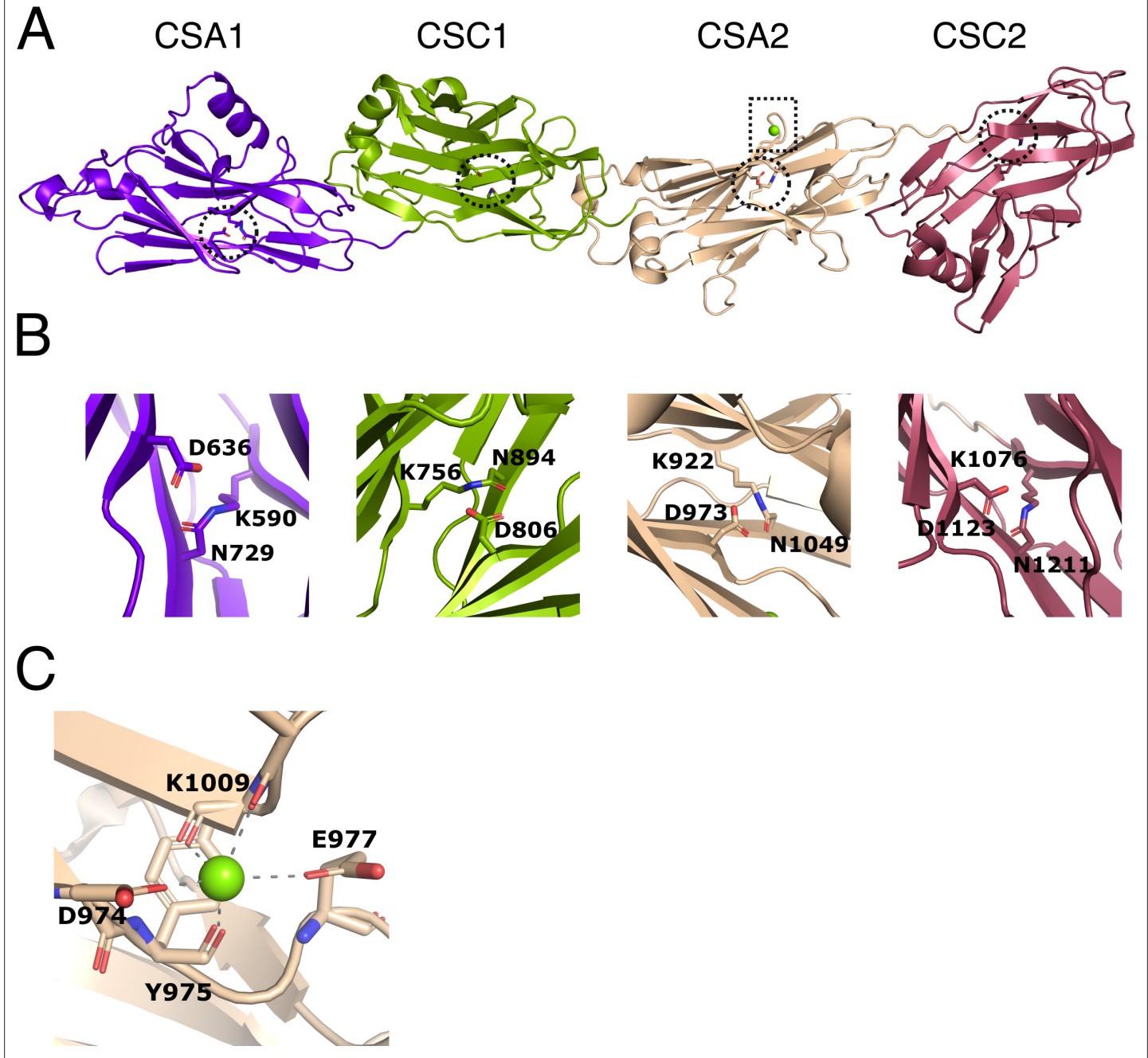

**Figure 2.** Structure of immunoglobulin (Ig)-like domains of PrgB. (**A**) The Ig-like domains are arranged as tandem pairs (CSA1–CSC1 and CSA2–CSC2). (**B**) Each Ig-like domain has an internal isopeptide bond (indicated by the striped circle in panel A) strengthening the structural integrity of the domain. Each of the four isopeptide bonds is between a lysine and an asparagine and further stabilized by an aspartic acid residue (highlighted residues are shown in stick representation). (**C**) A conserved metal binding site is situated in the CSA2 domain (highlighted by a striped box in panel **A**), here modeled with a $Mg^{2+}$ (green sphere).

The online version of this article includes the following source data and figure supplement(s) for figure 2:

**Figure supplement 1.** Structure of PrgB compared to the C-terminal domain of Pas (gray) (PDB code: 6E3F).

**Figure supplement 2.** Cryo-EM data processing scheme.

**Figure supplement 3.** Crystal structures of the PrgB polymer adhesin domain and immunoglobulin (Ig)-like domains in cartoon representation docked into the volumes acquired by single particle cryo-EM.

**Figure supplement 4.** Analysis of potential interaction of the polymer adhesin domain and stalk-like domains of PrgB.

*Figure 2 continued on next page*

*Figure 2 continued*

**Figure supplement 4—source data 1.** Raw image of the SDS-PAGE in *Figure 2—figure supplement 4B*.

**Figure supplement 5.** AlphaFold model of PrgB.

As expected (*Bhatty et al., 2015*), deletion of *prgB* also leads to a large decrease in biofilm formation (*Figure 3B*). In line with our observations from the aggregation assays, PrgB with a deletion of either the COI domain or more than a single Ig-like domain could not rescue the *E. faecalis* OG1RF pCF10Δ*prgB* biofilm formation phenotype. Only exogenous expression of PrgB$_{\Delta CSC2}$ can rescue the level of biofilm formation, but only to the level found in OG1RF pCF10, not to the level of exogenously expressed wild-type PrgB (*Figure 3B*, columns 1, 3, and 9). Intriguingly, the expression of exogenous PrgB$_{\Delta CSA1}$ in the OG1RF pCF10Δ*prgB* background did not restore biofilm formation, while it did restore the aggregation phenotype (*Figure 3A, B*, column 8).

The PrgB variants that failed to rescue the aggregation phenotype of the OG1RF pCF10Δ*prgB* strain (PrgB$_{\Delta COI}$, PrgB$_{\Delta CSA1-CSC2}$, PrgB$_{\Delta CSA1-CSC1}$, and PrgB$_{\Delta CSA2-CSC2}$) were also found to have a decreased plasmid transfer efficiency in the conjugation assay (*Figure 3C*, columns 4–7). However, PrgB$_{\Delta CSA1}$ and PrgB$_{\Delta CSC2}$ could only partially rescue the conjugation rate of the OG1RF pCF10Δ*prgB* strain (*Figure 3C*, columns 8 and 9), while they did rescue the aggregation phenotype.

To determine whether all PrgB variants were properly expressed, translocated and linked to the cell wall, we probed the protein levels in the cell wall fraction by western blot after 1-hr induction (corresponding to a time point relevant for the conjugation assay) and after overnight incubation (corresponding to a time point relevant to the aggregation and biofilm assays). All PrgB variants were present in the cell wall extract after 1-hr induction (*Figure 3—figure supplement 1A*), indicating that the production and folding of them are normal. However, the protein levels of PrgB$_{\Delta COI}$, PrgB$_{\Delta CSA1-CSC2}$, PrgB$_{\Delta CSA1-CSC1}$, and PrgB$_{\Delta CSA2-CSC2}$ were reduced after overnight incubation, as compared to wild type (*Figure 3—figure supplement 1B*, lanes 5–8 compared to lane 3). This indicates that the protein stability is decreased when these domains are missing, which could explain the observed aggregation deficiency after overnight incubation. Intriguingly, even though PrgB$_{\Delta CSA1}$ and PrgB$_{\Delta CSC2}$ are relatively stable (*Figure 3—figure supplement 1B*, lanes 9–10) and complemented the aggregation phenotype, they still could not fully rescue the biofilm phenotype. This suggests a functional loss of in these two variants.

## The conserved binding cleft in the polymer adhesin domain is essential for conjugation and biofilm formation, but not for aggregation

To investigate the role of the binding cleft in the polymer adhesin domain, we introduced single, double, and triple mutations to alter its conserved residues. The resulting *prgB* mutants were exogenously expressed in the background of pCF10Δ*prgB* for functional complementation; or in the background of wild-type pCF10 to test for any potential dominant negative effects as previously observed for PrgB$_{\Delta 246-558}$ (deletion of the polymer adhesin domain) (*Schmitt et al., 2018*). Notably, all the resulting PrgB variants fully restored the aggregation phenotype of pCF10Δ*prgB* (*Figure 4A*), but did not rescue the defective biofilm formation, nor the reduced conjugation efficiency (*Figure 4B, C*, columns 2–6). No dominant negative effects on aggregation or biofilm formation could be observed when these variants were expressed in the wild-type pCF10 background (*Figure 4A, B*, columns 7–10), although slightly reduced conjugation rates were observed (*Figure 4C*, columns 7–10). We have previously shown that the polymer adhesin domain binds eDNA and that this binding correlates with both biofilm formation and conjugation efficiency. Therefore, we wanted to test whether eDNA binding was affected in these PrgB variants. To do so, we purified both wild-type PrgB$_{188-1235}$ and the double binding cleft variant PrgB$_{188-1233}$:S442A,N444A to compare their DNA-binding affinities via electrophoretic mobility shift assays (EMSA). The results indicate that the introduced changes in PrgB did not affect its ability to bind eDNA, as the DNA-binding affinities of PrgB$_{188-1233}$ and PrgB$_{188-1233}$:S442A,N444A were the same within experimental error (*Figure 4—figure supplement 1*).

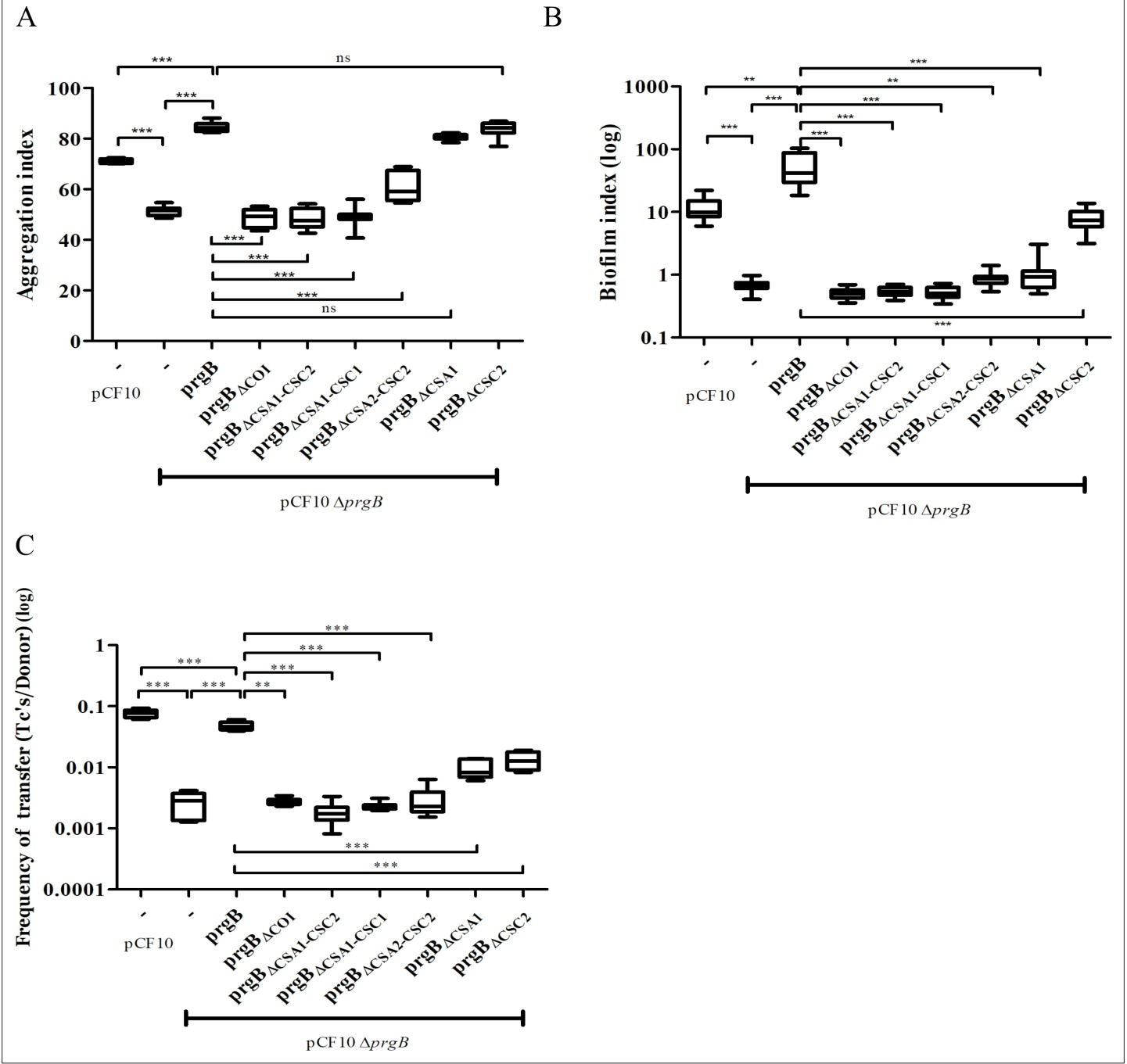

**Figure 3.** In vivo phenotypes of PrgB variants. PrgB variants are expressed from the pMSP3545S-MCS vector in the OG1RF pCF10Δ*prgB* background for phenotypic analysis with three assays: (**A**) cellular aggregation, (**B**) biofilm formation, and (**C**) conjugation assays. For all assays, OG1RF pCF10 carrying an empty vector serves as positive control and OG1RF pCF10Δ*prgB* with an empty vector as negative control. The value of each column represents the average of three independent experiments and the error bars represent the standard error of the mean (SEM). Statistical significance between the PrgB variants were analyzed with one-way analysis of variance, with * indicating $p < 0.05$, ** indicating $p < 0.01$, and *** indicating $p < 0.001$.

The online version of this article includes the following source data and figure supplement(s) for figure 3:

**Figure supplement 1.** Western blot showing the expression levels of the PrgB variants in *E. faecalis* cell wall extracts.

**Figure supplement 1—source data 1.** Raw images of SDS-PAGE.

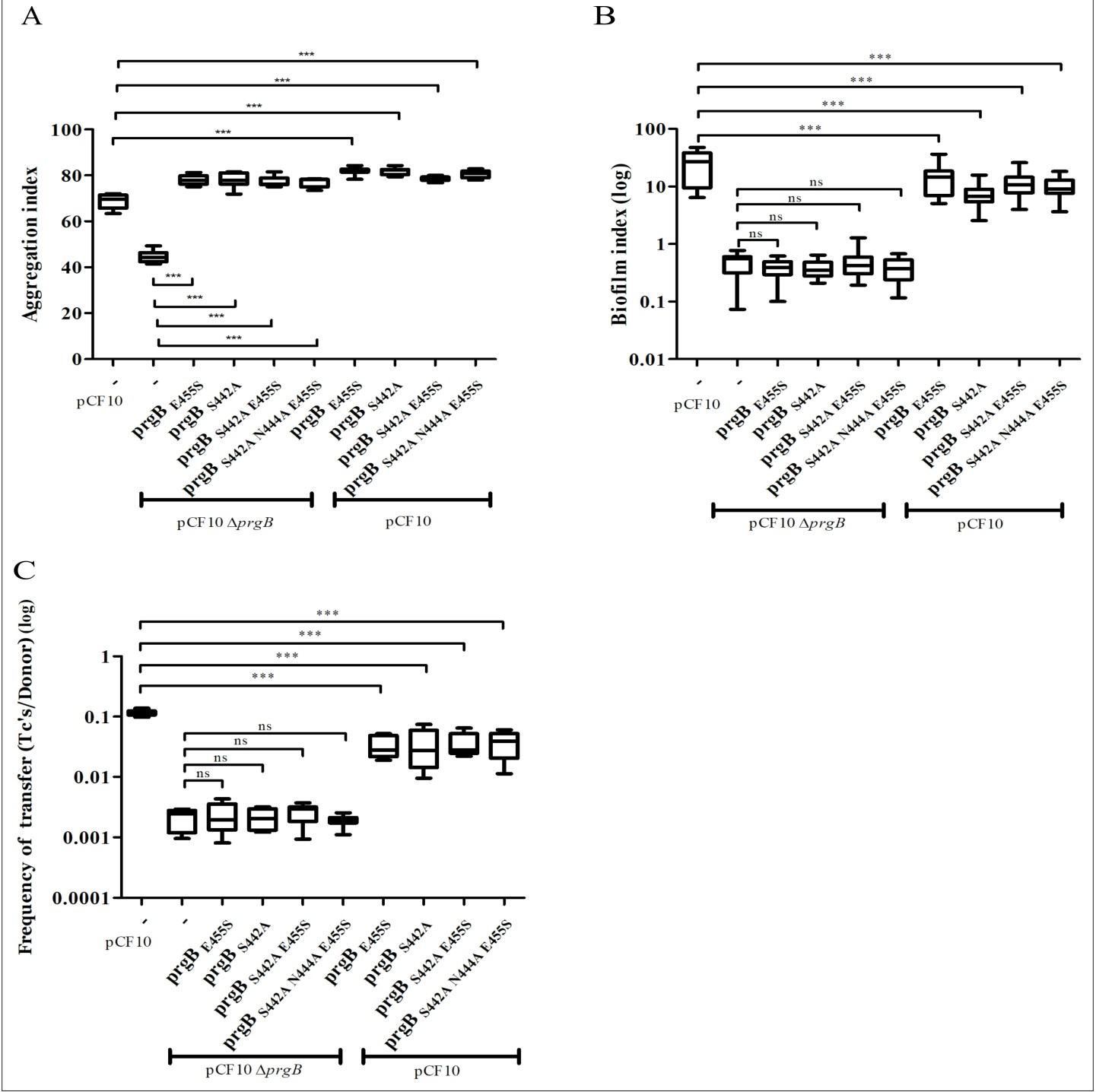

**Figure 4.** In vivo phenotypes of PrgB variants with point mutation(s) in the conserved site of the polymer adhesin domain. PrgB variants are expressed from the pMSP3545S-MCS vector in the OG1RF pCF10ΔprgB or OG1RF pCF10 strain and analyzed in: (**A**) cellular aggregation, (**B**) biofilm formation, and (**C**) conjugation assays. OG1RF pCF10 carrying the empty vector serves as positive control and OG1RF pCF10ΔprgB with the empty vector as negative control. The height of each column represents the average of three independent experiments and the error bars indicate the standard error of the mean (SEM). Statistical significance between the PrgB variants were analyzed with one-way analysis of variance, with * indicating $p < 0.05$, ** indicating $p < 0.01$, and *** indicating $p < 0.001$.

The online version of this article includes the following source data and figure supplement(s) for figure 4:

**Figure supplement 1.** Comparison of the DNA-binding affinity of the wild-type polymer adhesion domain from PrgB (PrgB$_{188–1234\ WT}$) and its S442A and N444A variant (PrgB$_{188–1234\ S442A\ N444A}$).

**Figure supplement 1—source data 1.** Raw tif files of the EMSAs in *Figure 4—figure supplement 1A*.

## Discussion

The presented data provide important insights for a widespread virulence factor in Gram-positive bacteria, since genes encoding PrgB homologs exist on a large number of conjugative plasmids (*Järvå et al., 2020*). In this study, we expand our structural knowledge of PrgB beyond the polymer adhesin domain, to now encompass almost the entire protein.

The crystal structure of PrgB$_{583-1233}$ shows that this part of PrgB consists of four tandemly arranged Ig-like domains. These Ig-like domains show a high degree of structural homology to Streptococcal surface proteins, usually found in the oral cavity (*Järvå et al., 2020*).

These homologous proteins have been indicated to bind various molecules, such as fimbria, collagen, and salivary agglutinin (SAG, also designated as glycoprotein 340), and these binding interactions have been shown to be vital for their function (*Forsgren et al., 2010*; *Mieher et al., 2021*; *Brady et al., 2010*; *Maddocks et al., 2011*; *Franklin et al., 2013*). However, we have not found any evidence that the Ig-like domains of PrgB bind to a specific substrate in our own experiments, nor have we found this in other reports. PrgB also does not contain a BAR domain, which is crucial for stable interactions between for example SspB from *S. gordonni* and Mfa-1 of *P. gingivalis* (*Purushotham and Deivanayagam, 2014*; *Daep et al., 2011*). However, since Ig-like domains are known to bind a large variety of ligands (*Halaby and Mornon, 1998*), we do not exclude the possibility that ligands for the Ig-like domains in PrgB will be found in the future. The only ligands that have been verified to interact with PrgB so far are eDNA and LTA, which have high affinity to the polymer adhesin domain (*Schmitt et al., 2018*).

The crystal structure of the Ig-like domains from PrgB, PrgB$_{583-1233}$ was complemented by single particle analysis of PrgB$_{188-1233}$ via cryo-EM (*Figure 2—figure supplement 3*). Despite the low resolution of the EM volumes, we tried to dock in the high-resolution crystal structures of the Ig-like domains and the polymer adhesin domain. The model of PrgB from cryo-EM indicated that the polymer adhesin domain might interact with the Ig-like domains in the absence of substrate, which could have implications for the function or regulation of PrgB. To test this hypothesis, we assayed the interaction between the polymer adhesin domain and the separately purified Ig-like stalk domain in vitro. The results did not show any interaction between these domains. However, in the full-length protein these two domains are attached via a linker region, which make their local concentration very high. Thus, we cannot exclude that these two domains can interact, but in that case the dissociation constant must be high (at least high µM range) and the interaction is thus unlikely to be physiologically relevant.

We have now obtained a structural basis to interpret almost all phenotypic data that is available for PrgB. Unfortunately, most of the mutants that were previously described did not correlate well with the newly determined domain boundaries. Therefore, we decided to create specific deletion mutants that were based on the new PrgB structure to determine the role of the various domains. At the N-terminus of PrgB, before the polymer adhesin domain, there is a predicted COI region (*Figure 1A*) that we wanted to investigate. To our surprise, the expression of PrgB$_{\Delta COI}$ could not rescue any of the aggregation, biofilm formation or conjugation phenotypes from a *prgB* deletion strain. However, in our experiments PrgB$_{\Delta COI}$ was unstable with substantially decreased amounts present in the cell-wall extract collected after overnight induction as compared to 1-hr induction (*Figure 3—figure supplement 1*), indicating that the COI region is important for protein production and/or stability. In line with this hypothesis, AlphaFold predicts this COI domain to interact with the linker between the polymer adhesin domain and the Ig-like domain (*Figure 2—figure supplement 5*). Since the linker region contains the sequence that PrgA recognizes for cleavage of PrgB (*Schmitt et al., 2020*), the lack of the COI domain and its potential shielding effect could explain the decreased stability of PrgB$_{\Delta COI}$. Deletion of all Ig-like domains (PrgB$_{\Delta CSA1-\Delta CSC2}$) renders the protein incapable to support aggregation, biofilm formation, and conjugation. However, exogenous expression of *prgB* with single Ig-like domain deletions, *prgB*$_{\Delta CSA1}$ and *prgB*$_{\Delta CSC2}$ in the pCF10Δ*prgB* background, restore cellular aggregation, while they do not rescue biofilm formation and conjugation (*Figure 3*). This was unexpected, as the polymer adhesin domain is predicted to mediate all the functions that were tested in these assays: aggregation, biofilm formation and conjugation (*Bhatty et al., 2015*; *Schmitt et al., 2018*). Our results, however, indicate that it is important to have all Ig-like domains present and properly folded. The cellular aggregation assays seem to indicate that PrgB can function when at least three Ig-like domains are present, as expression of both *prgB*$_{\Delta CSA1}$ and *prgB*$_{\Delta CSC2}$ can complement pCF10Δ*prgB*, but unfortunately this assay is not suitable to detect small changes. Based on the biofilm formation and

conjugation efficiency assays, which are more sensitive, we therefore conclude that even removing a single Ig-like domain strongly decreases the function of PrgB. We hypothesize that these Ig-like domains are required to present the polymer adhesin domain at the right distance from the cell.

The conserved Ser–Asn–Glu site in the negatively charged cleft of the polymer adhesin domain intrigued us, as its function is unknown. Any changes that we made in these conserved residues resulted in PrgB variants that did not facilitate biofilm formation or conjugation (*Figure 4*). Surprisingly the same PrgB variants did fully support cellular aggregation. This is thought-provoking, since various literature has shown that the PrgB functions in cellular aggregation, biofilm formation, and conjugation are strongly correlated. However, even a single-point mutation in this conserved site produced a PrgB variant that completely separates the cellular clumping phenotype from biofilm formation and conjugation. In vitro experiments showed that these point mutations did not impair PrgB binding to eDNA (*Figure 4—figure supplement 1*). A similar phenotypic pattern was observed with PrgB$_{\Delta CSA1}$ and PrgB$_{\Delta CSC2}$, which both could fully rescue aggregation but not biofilm formation. Our data therefore strongly indicate that PrgB has additional role(s), besides mediating cellular aggregation, to further support conjugation and biofilm formation. We therefore propose that PrgB performs at least one, currently unknown, additional function besides binding to eDNA and/or LTA from the cell wall of the recipient cell. It is highly likely that at least one of these additional functions is mediated by the conserved site in the polymer adhesin domain.

As described in the introduction, there is a plethora of *prgB* mutants made (see *Figure 5—figure supplement 1*) and phenotypically analyzed, predominantly by the group of Prof. Gary Dunny. In *Supplementary file 4*, we provide a summary of all *prgB* mutants that we have identified in the literature and our brief reinterpretation based on our new structural knowledge. Below, we will discuss a selected number of these mutants in detail.

As expected, most mutants that have an insertion or a mutation in the polymer adhesin domain (*Figure 5—figure supplement 1*) show a loss of protein function. Shortly, insertions at amino acids (a.a.) 358 and 359 are on the surface of the protein and likely leads to steric clashes that prevent the domain from binding to eDNA and LTA, whereas insertions at a.a. 439, 473, 517, and 546 are all in secondary structures that are central parts of the polymer adhesin domain and therefore are likely to disrupt folding of this domain.

The RGD motifs in PrgB were previously proposed to be involved in integrin binding and to promote adherence to human neutrophils, as well as internalization in cultured intestinal epithelial cells (*Vanek et al., 1999*; *Olmsted et al., 1994*). The new domain classification showed that these two RGD motifs

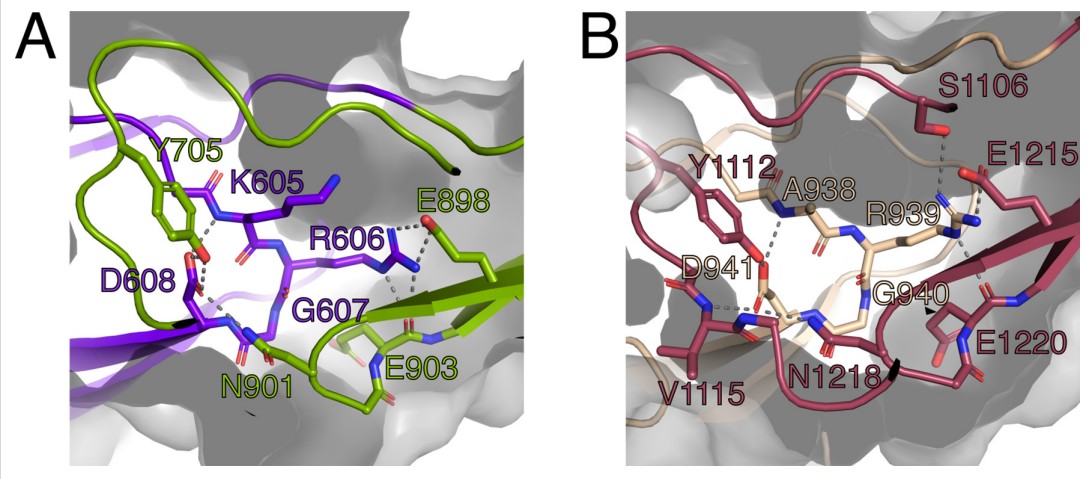

**Figure 5.** The RGD motifs play a role in the structural integrity of PrgB. (**A**) Close-up view of the RGD motif in the CSA1 domain (purple). Most of the important interactions, mainly hydrogen bonds, formed by this motif are to residues on the CSC1 domain (green). (**B**) Close-up view of the RGD motif in the CSA2 domain (sand colored). Most of the important interactions, mainly hydrogen bonds, formed by this motif are to residues on the CSC2 domain (dark red). Potential hydrogen bonds are marked by lines and the surface of the protein indicated by transparent gray.

The online version of this article includes the following figure supplement(s) for figure 5:

**Figure supplement 1.** Schematic overview of previously described *prgB* mutants.

are in CSA1 and CSA2, respectively (*Figure 1A*). The structure that we determined shows that these two sequence motifs are not surface exposed, but instead play an important role in the structural integrity of the interfaces between CSA1 and CSC1, and between CSA2 and CSC2 (*Figure 5*). Mutations in these motifs would thus very likely destabilize the folding of the tandem Ig-domains, which would explain the decreased PrgB biofilm formation observed in these strains (*Chuang et al., 2009*; *Chuang-Smith et al., 2010*). Thus, our new data strongly argue against the previously proposed direct binding interaction between the RGD sequences and integrins of host origin.

Deletion of residues 993–1138 leads to a loss of about half of both the CSA2 and CSC2 domains. Therefore, we were surprised to find that this mutant was described to behave like wild type in aggregation assays (*Waters et al., 2004*). Possibly the remaining parts of CSA2 and CSC2 form a (unfolded) linker region of similar length to the tandem CSA2–CSC2 structure, which allows PrgB to retain its function of promoting aggregation. Similarly, PrgB$_{\Delta668-1138}$, corresponding to a complete removal of CSC1 and CSA2 and approximately half of both CSA1 and CSC2 domains, could still support PrgB-mediated aggregation. Potentially the remaining residues (ca. 180 amino acids) could also form an unfolded linker region allowing for the variant to retain its aggregation phenotype. For PrgB$_{\Delta993-1138}$ and PrgB$_{\Delta668-1138}$, unfortunately no biofilm formation or conjugation efficiency were reported, but based on our results it is likely that those capabilities would have been severely compromised.

Taking past and present results into account combined with our new structural insights, we propose a new mechanistic model for the function of PrgB. Our in vivo data show that removal of one of the Ig-like domains does not largely affect PrgB function. Previous data also indicate that parts of the Ig-like domains can be deleted without affecting the function of PrgB. We therefore hypothesize that the Ig-like domains provide two important features to the protein. First, a rigid stalk that is needed to present the polymer adhesin domain at the correct distance from the cell wall. Second, providing the correct structural positioning for PrgA-mediated cleavage of the polymer adhesin domain, as the potential protease domain in PrgA is also presented away from the cell on a ~40-nm long stalk (*Schmitt et al., 2020*; *Järvå et al., 2020*). The exact distance of the polymer adhesin domain from the cell wall does not seem important for LTA binding, since aggregation can still take place even with partially disrupted Ig-like domains. However, the length of the stalk may be very important when it comes to facilitating both biofilm formation and conjugation. This indicates that effective mating-pair formation in conjugation may require a PrgB-mediated function distinct from pure aggregation, something that is further shown by the findings of the mutations in the conserved site in the polymer adhesin domain.

Besides providing a structural basis to explain about 30 years of work on PrgB, we here also uncovered that the conserved Ser–Asn–Glu site in the polymer adhesin domain likely provides additional functionality to PrgB that is needed for optimal biofilm formation and conjugation, but that does not affect cellular aggregation. To fully investigate the function of this conserved site in PrgB and other homologous virulence factors from Gram-positive bacteria, remains an exciting question to address in future research.

## Materials and methods
### Bacterial strains and growth conditions

See *Supplementary file 1* for a full list of all strains, plasmids, and oligonucleotides used. *Escherichia coli* Top10 was used in molecular cloning and grown in Lysogeny broth (LB). *E. coli* BL21 (DE3) was used for recombinant protein expression and grown in Terrific Broth (TB). The *E. faecalis* strains were cultured in Brain–Heart infusion broth (BHI) or Tryptic Soy broth without dextrose (TSB-D) as indicated in each assay. Concentrations of antibiotics for *E. coli* selection were as follows: ampicillin (100 μg/ml), kanamycin (50 μg/ml), spectinomycin (50 μg/ml), and erythromycin (150 μg/ml). In *E. faecalis* cultures, antibiotics were used as the following concentrations: tetracycline (10 μg/ml), fusidic acid (25 μg/ml), erythromycin (20 μg/ml for chromosome-encoded resistance; 100 μg/ml for plasmid-encoded resistance), spectinomycin (250 μg/ml for chromosome-encoded resistance; 1000 μg/ml for plasmid-encoded resistance), and streptomycin (1000 μg/ml). Plasmids were transformed to *E. coli* by heat-shock transformation, whereas *E. faecalis* strains were transformed by electroporation (*Bae et al., 2002*).

## Cloning and mutagenesis

To insert a multiple cloning site in pMSP3545S, DNA oligos of MCS_fwd and MCS_rev were resuspended in milliQ to 100 µM, mixed 1:1, denatured at 95°C for 15 min and slowly cooled to room temperature. The annealed MCS oligo was ligated into the gel-purified backbone fragment from pMSP3545S-*prgK* vector (*Laverde Gomez et al., 2014*) digested with NcoI and XbaI (removing the *prgK* insert). This was done in a 30 µl ligation mixture with 90 ng of the digested vector and a nine times molar access of insert. 5 µl of this ligation mixture was transformed into Top10 competent cells and plated on LB agar plates with spectinomycin (50 µg/ml) and erythromycin (150 µg/ml). The constructed plasmid was analyzed by restriction digestion and sequenced to confirm the correct insertion of the multiple cloning site and is further called pMSP3545S-MCS.

pMSP3545S-*prgB* was constructed by PCR of *prgB* from pCF10 with the NcoI-prgB-F and BamHI-stop-prgB-R primer pair and placement into the pMSP3545S-MCS vector via NcoI/BamHI restriction, gel purification of the correct DNA fragments and subsequent ligation. The constructed pMSP3545S-*prgB* was subsequently used as a template for mutagenesis creating pMSP3545S-*prgB* deletion or point mutation variants. This was carried out with inverse PCR (iPCR) using partially overlapping primer pairs. The iPCR products were gel purified with the DNA clean-up kit and digested with DpnI to remove residual template plasmid DNA. The processed iPCR products were then transformed to Top10 competent cells. For overexpression and protein purification, pET-$prgB_{246-558}$ and pET-$prgB_{188-1233}$ were transformed into *E. coli* BL21 (DE3) as previously described (*Schmitt et al., 2020*). $prgB_{188-1233}$ and $prgB_{580-1233}$ DNA fragments were amplified with primers mentioned in Supplementry File 1, and cloned into pINIT vector and then subcloned into the p7XC3GH vector via the FX cloning system (*Geertsma and Dutzler, 2011*). Mutations were introduced to p7XC3GH-$prgB_{188-1233}$ with the same iPCR approach to obtain the derivative plasmid of $prgB_{188-1233: S442A, N444A}$. All constructed plasmids were screened by PCR and verified by sequencing.

## Protein purification and crystallization

PrgB was produced as previously described (*Schmitt et al., 2018*). Briefly, $PrgB_{188-1233}$ was expressed with an N-terminal hexa-histidine tag from pET-$prgB_{188-1233}$ or a C-terminal deca-histidine and GFP (Green Fluorescent Protein) tag from p7XC3GH-*prgB* in *E. coli* BL21 (DE3). The cells were grown at 37°C in TB medium until they reached an $OD_{600}$ of 1.5. Then the temperature was lowered to 18°C and protein production was induced by adding 0.5 mM Isopropyl ß-D-1-thiogalactopyranoside (IPTG). Cells were grown for 16 hr before harvesting by centrifugation. Cells were resuspended in 20 mM HEPES/NaOH (pH 7.0), 300 mM NaCl, 30 mM imidazole, and 0.02 mg/ml DNase I and broken with a Constant cell disruptor at 4°C and 25 kPsi (Constant Systems). The cell lysate was clarified by centrifugation for 30 min at 30,000 × *g*, 4°C and incubated at 4°C with Ni-NTA (Macherey-Nagel). The Ni-NTA column was washed with 10 column volumes of 20 mM HEPES/NaOH (pH 7.0), 300 mM NaCl, 30–50 mM imidazole, and bound proteins were eluted from the column with the same buffer supplemented with 500 mM imidazole. The histidine affinity tags and the GFP when present were cleaved off from the purified protein fractions by incubation with TEV protease (for pET-*prgB*) or Prescission protease (for p7XC3GH-*prgB*) in a 1:100 ratio for 20 hr at 4°C. The cleaved proteins were loaded on a Superdex-200 Increase 10/300 GL column (Cytiva) equilibrated in 20 mM HEPES/NaOH (pH 7.0) and 150 mM NaCl. The elution profile showed two peaks corresponding to a PrgB dimer and monomer, as previously reported (*Schmitt et al., 2018*). These two peak fractions were pooled separately and concentrated on an Amicon Ultra Centrifugal Filter with a 30-kDa cutoff (Merck-Millipore). 10% glycerol was added to the concentrated protein fractions, which were subsequently flash frozen in liquid nitrogen and stored at −80°C.

## Native gel electrophoresis

Elution fractions from size-exclusion chromatography were mixed with native gel sample buffer (Invitrogen) and loaded on a Novex 4–20% Tris-Glycine gel (Invitrogen). Staining was carried out with InstantBlue Protein Stain (VWR).

## Structure determination via X-ray crystallography

Purified $PrgB_{188-1233}$ from the monomeric fraction, with a protein concentration of 15 mg/ml, were thawed and used in crystallization trials. Crystals of $PrgB_{188-1233}$ were grown in 8–12 weeks, at 20°C

by sitting drop vapor diffusion in a condition with 0.2 M CaCl$_2$ and 20% PEG 3350 and a protein to reservoir ratio of 1:1 in the drop. Crystals were flash cooled in liquid nitrogen. X-ray diffraction data of PrgB$_{188-1233}$ were collected on ID23-1 at the ESRF, France. The data were processed using XDS (*Kabsch, 2010*). The PrgB$_{188-1234}$ crystals belong to the monoclinic space group P2$_1$2$_1$2$_1$ and contain two molecules in the asymmetric unit. The crystallographic phase problem was solved using molecular replacement using PHASER, using the SspB homology model of the Ig-domains as search models (PDB: 2WOY) (*Forsgren et al., 2010*). Further building of the model was conducted in COOT (*Emsley et al., 2010*). The structure was refined to 1.85 Å with crystallographic R$_{work}$ and R$_{free}$ values of 20.9/24.6 using Refmac5 and PHENIX refine (*Winn et al., 2001*; *Adams et al., 2002*). The final PrgB$_{188-1233}$ model consists of residues 584–1234, and was validated using MolProbity (*Chen et al., 2010*). Atomic coordinates and structure factors have been deposited in the Protein Data Bank (PDB code: 8BEG).

## Sample preparation for electron microscopy

PrgB$_{188-1233}$ dimer fractions were thawed on ice and loaded on a Superdex 200 10/300 GL gel filtration column (GE Healthcare) equilibrated in 20 mM HEPES/NaOH pH 7.0 and 150 mM NaCl. Protein peak fractions, corresponding to the dimer, were diluted to 0.1–0.3 mg/ml and for the DNA-bound structures 120-bp ssDNA (*Table 1*) was added in 1:1.2 molar ratio (protein:DNA) and samples were incubated for 15 min on ice. For both apo and DNA-bound samples, 4 µl of sample was applied to glow discharged Quantifoil 300 mesh 1.2/1.3 (Quantifoil) grids at 4°C and 90–100% humidity, blotted for 1 s with blot force −5, and plunge-frozen in liquid ethane using a Vitrobot Mark IV (Thermo Fisher Scientific).

## Cryo-EM data collection

Cryo-EM data were collected on an FEI Titan Krios transmission electron microscope (Thermo Fisher Scientific), operated at 300 keV that was equipped with a K2 direct electron detector. Data were collected by the AFIS method using the EPU software V2.8.0 (Thermo Scientific) at a nominal magnification of ×165,000 (0.82 Å pixel size). Data collection parameters are listed in *Supplementary file 2*, and the general workflow showing representative micrographs, 2D and 3D classes are shown in *Figure 2—figure supplement 2*. A total number of 2787 movie stacks were collected for apo PrgB and 1670 for DNA-bound PrgB.

**Table 1.** X-ray data collection and refinement statistics.

Values within parenthesis correspond to the highest resolution shell.

| Data collection summary | PrgB$_{584-1233}$ |
|---|---|
| Space group | P2$_1$2$_1$2$_1$ |
| Cell dimensions | |
| a, b, c (Å) | 76.0, 87.5, 224.8 |
| α, β, γ (°) | 90, 90, 90 |
| Resolution (Å) | 47.3–1.84 (1.91–1.84) |
| Completeness (%) | 99.6 (99.5) |
| R$_{meas}$ (%) | 0.03 (0.99) |
| I/σ (I) | 12.1 (1.0) |
| CC(1/2) | 1 (0.67) |
| Redundancy | 2.0 (2.0) |
| No. unique reflections | 130,301 (12,877) |
| **Refinement summary** | |
| Resolution (Å) | 47.3–1.84 |
| R$_{work}$ (%) | 20.9 |
| R$_{free}$ (%) | 24.5 |
| Number of atoms | |
| Protein | 10,080 |
| Water | 816 |
| Ther ligands | 2 |
| *B*-factors | |
| Protein | 44.9 |
| Water | 47.7 |
| Ther ligands | 48.3 |
| r.m.s. deviations | |
| Bond lengths (Å) | 0.006 |
| Bond angles (°) | 0.79 |
| Ramachandran statistics | |
| Utliers (%) | 0 |
| Allowed (%) | 1.5 |
| Favored (%) | 98.5 |

## Cryo-EM data processing

Cryo-EM data of apo PrgB and DNA-bound PrgB were processed in the same way, but separately using cryoSPARC (v3.2.0-3.3.1) (*Punjani et al., 2017*). Beam-induced motion was corrected using standard settings, where start frame 1 was excluded, followed by per-micrograph contrast transfer function estimation. For apo PrgB a subset of 500 micrographs were picked using the blob picking tool with a 100–200 Å particle diameter, followed by extraction of 170,826 particles with a box size of 384 pix. Picked particles were subjected to consecutive rounds of 2D classifications. Subsequently, representative 2D classes were selected as input for picking of the full dataset, using the template picker tool. PrgB with DNA was directly picked using blob picker with a 100–300 Å particle diameter and standard settings and extracted with 384 pix. PrgB without and with DNA were then separately subjected to 2D classifications resulting in a final number of 283,630 and 163,566 particles, respectively. Particles from selected classes were combined and used in ab initio reconstruction. The initial volume was then subjected to homogenous 3D refinement and the resolution was calculated using the gold standard Fourier shell correlation (FSC threshold, 0.143) and found to be 8 and 11 Å for the apo and DNA-bound structure, respectively. The volumes of apo and DNA-bound PrgB have been deposited in the Electron Microscopy Data Bank (EMDB codes: EMD-16001 and EMD-16002).

The adhesion domain (PDB code: 6EVU) (*Schmitt et al., 2018*) and stalk domain (PDB code: 8BEG) were initially docked into the EM volume using Chimera (*Pettersen et al., 2004*) and subsequently run through Namdinator (*Kidmose et al., 2019*) using 10 Å resolution and standard settings. The output was then fitted in the EM volume in Chimera (v 1.15rc, *Pettersen et al., 2004*), where figures also were generated.

## Aggregation assay

*E. faecalis* strains were inoculated in BHI medium with the indicated antibiotics and cultured overnight at 37°C. Overnight cultures were diluted in a 1:100 ratio in TSB-D with the indicated antibiotics, 10 ng/ml cCF10, and 50 ng/ml nisin and dispensed into polystyrene cuvettes (Sarstedt) in 0.9 ml triplicates. These were incubated for 24 hr at 37°C without agitation. Afterwards, the optical density of each sample was determined at 600 nm both before ($OD_{sup}$) and after ($OD_{mix}$) vigorously mixing of the bacterial culture by pipetting. The autoaggregation percentage was then calculated as follows: $100 \times [1 - (OD_{sup}/OD_{mix})]$ (*Bhatty et al., 2015*; *Waters and Dunny, 2001*).

## Biofilm assay

*E. faecalis* strains were inoculated in BHI with the indicated antibiotics and kept 16 hr at 37°C. The next morning, they were diluted in a 1:100 ratio in TSB-D with the indicated antibiotics, 10 ng/ml cCF10, and 50 ng/ml nisin. 200 μl fractions were dispensed into a 96-well microtiter plate (Costar) with 8 replicates per strain. 200 μl TSB-D fractions were used as blanks. The 96-well plate was then incubated aerobically at 37°C without agitation in a humidified chamber for 24 hr. The suspension was transferred to another 96-well plate to determine the optical density at 600 nm ($OD_{600}$). The plate containing the biofilm was washed with distilled water three times and then left to air dry at room temp for 2.5 hr. The biofilm was stained with 100 μl 0.1% (wt/vol) safranin (Sigma) at room temp for 20 min, then washed three times with distilled water and left to air dry at room temperature. Afterwards the absorbance was determined using a plate reader (BMG Labtech) at 450 nm. Biofilm production was calculated as an index of safranin staining of the cell biomass divided by absorbance of its optical density ($OD_{450}/OD_{600}$) (*Willett et al., 2019*).

## Analysis of the protein levels of PrgB variants in cell wall extracts

For the 1-hr time point, samples of each strain were diluted in a 1:25 ratio in TSB-D with the indicated antibiotics, cultured at 37°C for 2 hr, until optical density 0.6 was reached, and then induced with 10 ng/ml cCF10, and 50 ng/ml nisin for 1 hr. Overnight samples were harvested after induction and incubation overnight as described in the aggregation assay. The cell pellets were treated with lysozyme buffer (10 mM Tris, pH 8.0, 1 mM Ethylenediaminetetraacetic acid (EDTA), 25% sucrose, 15 mg/ml lysozyme) for 30 min at 37°C. The lysozyme-treated bacterial cells were spun down at 13,000 × *g*, 4°C for 5 min. The supernatant, containing the cell wall extract, was mixed with protein loading dye and boiled at 100°C for 12 min. Subsequently, the samples were run on a 8% sodium dodecyl

sulfate–PAGE, transferred to western blot and probed with the PrgB antibody produced in rabbit (*Christie et al., 1988*; *Chen et al., 2007*; *Chen et al., 2008*).

## Conjugation assay

Donor (OG1RF pCF10 pMSP3545S derivative strains) and recipient (OG1ES) strains were inoculated in BHI with the indicated antibiotics and incubated overnight at 37°C with agitation. Overnight cultured strains were refreshed in BHI without antibiotics in a 1:10 ratio, and donor strains were induced with 50 ng/ml nisin (Sigma). All strains were then incubated at 37°C for 1 hr without agitation. Afterwards each of the donor strains was mixed with the recipient cells in ratio of 1:10 and mated at 37°C statically for 3.5 hr. These mixtures were then serially diluted with BHI and plated out in triplicates on BHI agar plates supplemented with tetracycline and spectinomycin (to select for donor cells), or with tetracycline, erythromycin, and streptomycin (to select for transconjugants). Plates were incubated at 37°C for 48 hr, counted and enumerated for colony-forming units (CFU). The plasmid transfer rate was determined as CFU of transconjugant over CFU of donor (Tc's/ Donors) (*Schmitt et al., 2018*).

## Electrophoretic mobility shift assay

EMSA was carried out in the same way as previously described (*Schmitt et al., 2018*). 0.1–20 µM PrgB (wild type and variants) were mixed with 50 nM 100 bp long double-stranded DNA (*Schmitt et al., 2018*). Samples were incubated for 1 hr at 20°C before loading them onto a 6% Tris-Borate, EDTA buffer (TBE)-based native acrylamide gel for electrophoresis for 90 min at 50 V and 6°C. Gels were subsequently stained with 3×GelRed (Biotium) in distilled water for 30 min and imaged with a Chemidoc system (Bio-Rad). Quantification of the DNA bands after imaging was done in ImageLab (Bio-Rad).

## Statistical analysis

All in vivo data are from three independent experiments and were plotted and analyzed using GraphPad Prism (version 5.0) (GraphPad Software). The indicated error is the standard deviation over three biologically independent replicates. Statistical significance between the PrgB variants were analyzed with one-way ANOVA analysis of variance, with * indicating $p < 0.05$, ** indicating $p < 0.01$, and *** indicating $p < 0.001$.

## Material availability

All data generated or analyzed during this study are included in this published article and its supplemental information. All structural data have been deposited in the Protein Data Bank (PDB) and the Electron Microscopy Data Base (EMDB) and is publicly available. DOIs are listed in the key resources table. Any additional information required to reanalyze the data reported in this paper, for example bacterial strains, is available from the corresponding author upon request.

## Acknowledgements

The authors would like to thank Prof. Gary Dunny for very fruitful discussions about the results and Dr. Karim Rafie and Annika Breidenstein for discussions about EM data processing. We acknowledge MAX IV Laboratory for time on Beamline BioMax under Proposal 20180236. Research conducted at MAX IV, a Swedish national user facility, is supported by the Swedish Research council under contract 2018-07152, the Swedish Governmental Agency for Innovation Systems under contract 2018-04969, and Formas under contract 2019-02496. We also acknowledge the synchrotrons Swiss Light Source (Paul Scherrer Institute, Switzerland) for time at beamline PX1 and the ESRF (France) for time at beamlines ID23 and ID30. The EM data were collected at the Umeå Core Facility for Electron Microscopy, a node of the Cryo-EM Swedish National Facility, funded by the Knut and Alice Wallenberg, Family Erling Persson and Kempe Foundations, SciLifeLab, Stockholm University, and Umeå University. This work was supported by grants from the Swedish Research Council (2016-03599), Knut and Alice Wallenberg Foundation, Kempestiftelserna (SMK-1762 and SMK-1869), and Carl-Tryggers stiftelse (CTS 18:39) to RPAB.

## Additional information

### Funding

| Funder | Grant reference number | Author |
|---|---|---|
| Vetenskapsrådet | 2016-03599 | Ronnie P-A Berntsson |
| Kempestiftelserna | SMK-1762 | Ronnie P-A Berntsson |
| Kempestiftelserna | SMK-1869 | Ronnie P-A Berntsson |
| Carl Tryggers Stiftelse för Vetenskaplig Forskning | CTS 18:39 | Ronnie P-A Berntsson |

The funders had no role in study design, data collection, and interpretation, or the decision to submit the work for publication.

### Author contributions

Wei-Sheng Sun, Conceptualization, Data curation, Formal analysis, Validation, Investigation, Visualization, Writing – original draft, Writing – review and editing; Lena Lassinantti, Investigation, Writing – original draft, Writing – review and editing; Michael Järvå, Conceptualization, Investigation, Writing – review and editing; Andreas Schmitt, Investigation, Writing – review and editing; Josy ter Beek, Supervision, Investigation, Writing – original draft, Writing – review and editing; Ronnie P-A Berntsson, Conceptualization, Supervision, Funding acquisition, Writing – original draft, Project administration, Writing – review and editing

### Author ORCIDs

Wei-Sheng Sun (iD) http://orcid.org/0000-0001-9738-8862
Lena Lassinantti (iD) http://orcid.org/0000-0001-5470-591X
Josy ter Beek (iD) http://orcid.org/0000-0003-4165-9277
Ronnie P-A Berntsson (iD) http://orcid.org/0000-0001-6848-322X

Reviewer #1 (Public Review): https://doi.org/10.7554/eLife.84427.3.sa1
Reviewer #2 (Public Review): https://doi.org/10.7554/eLife.84427.3.sa2
Author Response https://doi.org/10.7554/eLife.84427.3.sa3

## Additional files

### Supplementary files

- Supplementary file 1. Strains, plasmids, and oligonucleotides.
- Supplementary file 2. Cryo-EM data collection and refinement.
- Supplementary file 3. Top hits of Dali search based on PDB of PrgB Ig-like domains.
- Supplementary file 4. Structural interpretation of previously observed PrgB phenotypes.
- MDAR checklist

### Data availability

Diffraction data have been deposited in the PDB under the accession code 8BEG. EM volumes have been deposited in the EMDB under the accession codes EMD-16001 and EMD-16002.

The following datasets were generated:

| Author(s) | Year | Dataset title | Dataset URL | Database and Identifier |
|---|---|---|---|---|
| Sun W, Berntsson RPA | 2022 | Structure of Ig-like domains from PrgB | https://www.rcsb.org/structure/8BEG | RCSB Protein Data Bank, 8BEG |
| Lassinantti L, Berntsson RPA | 2022 | apo PrgB | https://www.ebi.ac.uk/emdb/EMD-16001 | Electron Microscopy Data Bank, EMD-16001 |
| Lassinantti L, Berntsson RPA | 2022 | PrgB with DNA | https://www.ebi.ac.uk/emdb/EMD-16002 | Electron Microscopy Data Bank, EMD-16002 |

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
