## [Editor Report · eLife assessment]

This study presents **valuable** structural data for the bacterial adhesin PrgB, an atypical microbial cell surface-anchored polypeptide that binds DNA. There is **convincing** support for the claims regarding the overall function and importance of individual domains, which integrate a wide range of new and previously published experimental data. The structure-based model of PrgB molecular activity will be impactful in the field of bacterial adhesins, conjugation, and biofilm formation, especially because it focuses on a clinically-important Gram-positive pathogen, whereas most work in the field has been focused on Gram-negative model systems.

---

## [Referee Report · Reviewer #1 (Public Review)]

Sun and colleagues outline structural and mechanistic studies of the bacterial adhesin PrgB, an atypical microbial cell surface-anchored polypeptide that binds DNA. The manuscript includes a crystal structure of the Ig-like domains of PrgB, cryo-EM structures of the majority of the intact polypeptide in DNA-bound and free forms, and an assessment of the phenotypes of *E. faecalis* strains expressing various PrgB mutants. Generally, the study has been conducted with a good level of rigor, and there is consistency in the findings. Initial concerns about inferences initially made from low-resolution Cryo-EM structures have been addressed experimentally and the manuscript correspondingly updated.

---

## [Referee Report · Reviewer #2 (Public Review)]

Having previously solved the X-ray crystallographic structure of the polymer adhesin domain (PAD) of PrgB from *E. faecalis*, the authors looked to build on that work by crystallizing a nearly full-length construct of PrgB. Though they were successful in their crystallization endeavors, the crystal contained only what was previously thought to be two domains with RGD motifs. The authors' high-resolution structure shows that in fact the C-terminal portion of PrgB is made up of four immunoglobulin-like domains. The authors then set out to collect single-particle cryoEM data in a bid to obtain a full-length structure of PrgB, both in the presence and absence of ssDNA. The authors were only able to obtain quite low-resolution data, which they fit their crystal structures into. The authors then used these structures to inform the design of novel deletion mutants and point mutations, as well as to rationalize years of phenotypic data from other published mutants.

The X-ray crystallographic structure is beautiful and in combination with their in vivo data allowed them to propose a model where PrgB positions cells at an appropriate distance for conjugation. The in vivo experiments appear to be done well and the authors' discovery that the Ser-Asn-Glu is not important for generalized aggregation but has an additional yet unknown role in conjugation and biofilm formation is exciting and well supported by their data.

[Editors' note: In response to reviews of a previous version of this manuscript, the authors have carried out additional experiments that have strengthened the already convincing aspects of the work. We commend the authors for responding to questions raised by the reviewers about the inference of interactions of in vivo importance inferred from low-resolution cryo-EM studies by carrying out and reporting on additional experiments that fail to confirm their initial speculative model. The current work is stronger and more convincing as a result.]

---

## [Author Response]

The following is the authors’ response to the original reviews.

First of all, we would like to again thank the reviewers for their work. We appreciate the constructive review comments and useful suggestions to further improve our article. With those comments in mind, we have now revised our manuscript. Please see below for a point-by-point response (our responses in green) to all comments.

**Reviewer #1 (Recommendations For The Authors):**
Sun and colleagues outline structural and mechanistic studies of the bacterial adhesin PrgB, an atypical microbial cell surface-anchored polypeptide that binds DNA. The manuscript includes a crystal structure of the Ig-like domains of PrgB, cryo-EM structures of the majority of the intact polypeptide in DNA-bound and free forms, and an assessment of the phenotypes of *E. faecalis* strains expressing various PrgB mutants.Generally, the study has been conducted with a good level of rigor, and there is consistency in the findings. However, I do have some specific technical concerns relating to the study that necessitate the undertaking of additional experiments. These are summarized as follows:1. Recombinant PrgB188-1233 produced in the study purifies as a mixture of monomeric and dimeric species separatable by SEC. There is very limited discussion in the text re. the significance and/or implications of this. Is it feasible that the dimeric form is biologically relevant in the context of the in vivo situation? Or alternatively, is this simply an artifact of protein production?

Experimental data that we published in 2018 indeed indicates that the dimer is relevant in the in vivo situation. We did not discuss this here since this was discussed in detail in the previous paper: Schmitt et al, 2018. We have now added a bit more information on this in the results section, highlighting this, so that it is clearer to the reader (lines 114-116).

1. The authors see no evidence of the adhesive domain of PrgB in their PX structure highlighting that this must have been cleaved during crystallisation. Is this claim supported by an inspection of the crystal packing? It could be that this region of the protein is dynamic within the context of the crystal and is thus not observed. This should be clarified in the text either way.

The crystal packing does not provide any space for the PAD. We have added this to the results section. We have added a sentence describing this in lines 122-124.

1. The Cryo-EM structures reported are both at ~10-angstrom resolution. Are the authors truly confident in the placement of their crystal structures on these maps? Visual inspection indicates that their positioning of the PrgB domains into the EM envelopes is somewhat questionable. The authors need to provide some quantitative measures of the quality of their domain fitting. The narrative of the manuscript very much hinges on this being correct.

This is something that the other reviewer also commented on. The fitting of the crystal structures in the maps are indeed not optimal, but was the best we could do with the available data. In line with point #6, we have now constructed new protein variants of the stalk domain (the four Ig-like domains) alone, and have assayed it’s interaction with the PAD in vitro using native gels and size exclusion chromatography. The outcome of these experiments is that the two domains do not interact in any substantial way on their own. Thus, the added experiments do not support the hypothesis that the PAD interacts with the Ig-like domains, at least not without the local high concentration provided by the linker region in the in vivo situation.

To account for these new experiments, we have moved the cryo-EM structure to the supplement, and rewritten this part of the manuscript to say that the cryo-EM data indicated that there might be an interaction, but that we have not been able to verify this in vitro, indicating that if the interaction at all exists it must have a low affinity and is likely not physiologically relevant. In line with this, we have also further modified the text throughout the manuscript to account for this.

1. The manuscript would be significantly strengthened if the authors could include confirmatory hydrodynamic data in support of the observed conformational reorganization of PrgB in the presence of DNA. SAXS analysis of the DNA-free and bound complexes would be ideal for this and would also help address the issues raised above in pt 3.

To analyze PrgB radius with and without DNA, we tried both SEC-MALS and DLS experiments. It proved difficult to obtain precise and reproducible values, but the initial data indicated that no large changes were observed upon DNA binding. As we could also not measure specific interaction between the PAD and the stalk in vitro, we did not perform SAXS experiments. As mentioned in the response to point #3, we have modified the results and discussion regarding the potential interaction of th PAD and Stalk domains.

1. The authors present binding studies of various PrgB mutant-expressing strains. A number of the mutations generated delete significant portions of the polypeptide. Can the authors confirm that these mutant proteins are correctly folded despite the introduced mutations? It could be that loss of function is simply a consequence of mutation-induced misfolding. I would like to see some confirmatory data (CD, SEC, etc.) in support of the foldedness of the mutant proteins.

We cannot completely rule out that the folding of some of the variants is affected in *E. faecalis*. However, CD or SEC experiments would only give indications of the contrary if the overall fold had been majorly affected in an in vitro situation where the protein is not anchored to the *E. faecalis* cell wall.

To alleviate this valid concern, we probed if all variants are correctly exported and linked to the cell-wall. Therefore we have now extracted the cell wall of *E. faecalis* producing wild-type or variant PrgB and performed Western blot . The results of the Western blot with cell wall extract largely matches the whole cell experiments that were in the initial manuscript. If a protein variant was largely misfolded, it would likely not be targeted and linked to the cell-wall, nor would it be stable in vivo. We have added this new data as a new fig 3 – figure supplement 1 and on lines 201-214

1. The authors suggest a direct interaction between the PAD and the stalk domains in PrgB. The discussion of this is very generic and no evidence to support this is provided other than the 10-angstrom resolution EM map. If they believe this to be the case, then additional evidence should be provided.

Answer: As mentioned previously, we have now performed additional in vitro experiments to probe this potential interaction, but conclude that this indication from the EM data is likely not a real high affinity interaction. In line with this, we have modified the results and discussion regarding this point, see also response to point #3 and 4.

**Reviewer #2 (Recommendations For The Authors):**
As currently presented, I don't feel that the cryoEM data support the authors' proposed model, largely because the fit of the crystal structures to the EM volumes does not seem entirely reasonable for the apo- dataset and because the EM volume for the ssDNA bound dataset is not even contiguous. For me to believe the model as it is currently built, I would want to see a dataset with the PAD deleted, showing that its proposed density disappears, or a dataset with a PAD-specific antibody as a fiducial marker. It would be nice to see some goodness of fit metric with a comparison to other crystal structures fit such low-resolution data as well. At the very least, the authors must include the standard cryoEM workflow supplementary figure showing representative micrographs, 2Ds, and 3Ds along with particle numbers.

In line with the comments raised by reviewer #1, we have now added more experiments where we have analyzed the potential interaction between PAD and the stalk domain. From this new data, it looks like they do not interact with any substantial affinity, at least not on their own without any linker region holding them together, and that this interaction if it all exist likely is not physiologically relevant. The cryo-EM data has been moved to the supplement as we agree with both reviewers that the resolution, and the fitted model, is not good enough to draw any hard conclusions. The standard table for the cryoEM workflow was present as supplementary table 2, where eg particle numbers etc are described, but we have now also added a new supplementary fig 2 – figure supplement 2 that shows the EM processing workflow, including representative micrographs, 2D and 3D classes. We debated whether we should remove the EM data, but decided against it in line of transparency and to explain why the interaction studies with the PAD and stalk domains were performed.

The X-ray crystallographic structure is very nice, but I was a bit surprised by the R factors in Table 1. After downloading the structure factors and coordinates from the PDB (thank you for depositing before submission!) I was able to see quite a few positive peaks in the difference map that could probably use some cleaning up. I realize I may just be a bit of a masochist when it comes to adding/deleting waters and moving around side chains to get things just right, but for such lovely data, I would have liked to see the model polished up a bit more. I was going to say that the isopeptide bond should be modelled, but I can see from a cursory Google that the authors did in fact try to find a way to model this and that it is indeed a bit of a pain.

The model refinement proved surprisingly recalcitrant with regards to the remaining difference density, so we took the decision to only model what was solidly there (which leads to slightly higher R factors). We did indeed try to model the isopeptide bond, but we did not find a good way to do so (despite trying quite extensively), and ended up determining them as a linker in the PDB file, so that the bond shows up when one opens the structure in eg. Pymol.

For protein production/purification in general I would have liked to see actual traces for the gel filtration and pure protein on a gel in a supplementary figure. I strongly believe that this type of information is so critical for future researchers looking to replicate or build upon published work so that they have some sense that what they are doing is working in the way it should be.

We have now added a supplementary figure (as new Fig. 1 – figure supplement 1) that shows SEC and SDS-PAGE for the purification of PrgB188-1233.

Finally, I think for the in vivo data it only makes sense to show the reader whether any or all the differences measured across your different mutants are statistically significant. Having done the graphing and analysis in GraphPad this should be a simple thing to achieve.

We have now added statistical test (One way Anova) that show the statistical significance between the mutants, and show that in Fig 3 and Fig 4.

Overall, I think it's a very nice paper and while I feel that the cryoEM data in its current form doesn't support the model of occlusion from PrgA, I also don't think that removing the cryoEM data and that specific mechanistic idea from the paper detracts from its overall message and impact.

Thank you for those comments.